# Effects of Wine and Tyrosol on the Lipid Metabolic Profile of Subjects at Risk of Cardiovascular Disease: Potential Cardioprotective Role of Ceramides

**DOI:** 10.3390/antiox10111679

**Published:** 2021-10-25

**Authors:** Jose Rodríguez-Morató, Anna Boronat, Gabriele Serreli, Laura Enríquez, Alex Gomez-Gomez, Oscar J. Pozo, Montserrat Fitó, Rafael de la Torre

**Affiliations:** 1Integrative Pharmacology and Systems Neuroscience Research Group, IMIM (Hospital del Mar Medical Research Institute), Dr. Aiguader 88, 08003 Barcelona, Spain; jose.rodriguez@upf.edu (J.R.-M.); aboronat@imim.es (A.B.); laura.enriquez03@estudiant.upf.edu (L.E.); 2Department of Experimental and Health Sciences, Universitat Pompeu Fabra (CEXS-UPF), Dr. Aiguader 80, 08003 Barcelona, Spain; agomez@imim.es (A.G.-G.); opozo@imim.es (O.J.P.); 3Spanish Biomedical Research Centre in Physiopathology of Obesity and Nutrition (CIBEROBN), Instituto Salud Carlos III, 28029 Madrid, Spain; mfito@imim.es; 4Department of Biomedical Sciences, University of Cagliari, Cittadella Universitaria SS 554, 09042 Monserrato, Italy; gabriele.serreli@unica.it; 5Applied Metabolomics Research Group, IMIM (Hospital del Mar Medical Research Institute), Dr. Aiguader 88, 08003 Barcelona, Spain; 6Epidemiology Program, Cardiovascular Risk and Nutrition Research Group, IMIM (Hospital del Mar Medical Research Institute), Dr. Aiguader 88, 08003 Barcelona, Spain

**Keywords:** cardiovascular disease, ceramides, ethanol, randomized clinical trial, tyrosol, white wine

## Abstract

Ceramides are a class of sphingolipids which have recently been shown to be better cardiovascular disease (CVD) risk predictors than traditional CVD risk biomarkers. Tyrosol (TYR) is a dietary phenolic compound known to possess cardioprotective effects per se or through its in vivo active metabolite hydroxytyrosol. The purpose of this study was to evaluate the effects of the co-administration of white wine (WW) and TYR on circulating levels of ceramides and other lipids in humans at high CVD risk. Volunteers underwent a randomized controlled crossover clinical trial (4-week duration per intervention) with three different interventions: control, WW, and WW enriched with a capsule of TYR (WW + TYR). Endothelial function cardiovascular biomarkers and plasma lipidomic profile were assessed before and after each intervention. It was found that the WW + TYR intervention resulted in lower levels of three ceramide ratios, associated with an improvement of endothelial function (Cer C16:0/Cer C24:0, Cer C18:0/Cer C24:0, and Cer C24:1/Cer C24:0), when compared to the control intervention. Moreover, WW + TYR was able to minimize the alterations in plasma diacylglycerols concentrations observed following WW. Overall, the results obtained show that the antioxidant TYR administered with WW exerts beneficial effects at the cardiovascular level, in part by modulating blood lipid profile.

## 1. Introduction

Cardiovascular diseases (CVDs) are the leading cause of death globally. According to the WHO, they take around 17.9 million lives each year, representing 32% of all global deaths [1]. Classic CVD risk factors (i.e., hypertension, dyslipidemia, diabetes, and smoking) play a central role in the prevention of CVD. However, classic CVD risk factors have limitations, as they fail to identify all patients at risk [2]. For instance, approximately 50% of the people developing coronary heart disease have been classified as having low or intermediate risk based on current risk algorithms [3]. This fact has driven important efforts to find alternative new biomarkers to improve prevention, prediction, diagnosis, and prognosis of CVD.

Ceramides are a class of circulating sphingolipids that have received increasing attention due to their potential as CVD biomarkers [4] as they have been shown to better predict CVD risk irrespective of traditional risk factors [5]. Several studies in humans indicate that circulating ceramides are not only accurate biomarkers of adverse CVD outcomes but also drivers of CVD [6]. Ceramides are bioactive lipids which act as second messengers and play key roles in inflammation, cell signaling, adhesion, migration, differentiation, angiogenesis, and proliferation [4]. They are intermediaries between insulin resistance and cardiometabolic diseases, and are induced by overnutrition and inflammatory cytokines [7]. Ceramides are endogenously synthesized and circulating concentrations of ceramides are altered by some drugs and lifestyle habits (i.e., diet and exercise) [8].

Although a great variety of ceramides exist, the four most studied are the long-chain ceramides C16:0 and C18:0, as well as the very-long-chain ceramides C24:1 and C24:0 (for nomenclature purposes, we follow the naming CXX:Y, where XX refers to the number of carbon atoms and Y indicates the number of double bonds). Given that ceramide C24:0 has an opposite behavior than the other ceramides, its use in ratios with other ceramides is becoming common (i.e., Cer C16:0/Cer C24:0, Cer C18:0/Cer C24:0, and Cer C24:1/Cer C24:0) [7,8,9,10,11]. According to recent reports, these ceramide ratios are significant predictors for cardiovascular (CV) death in individuals with coronary artery disease [7,10,11] and are associated with the risk of major CV events in healthy individuals [11]. Importantly, when compared to single ceramide values or the traditional lipid biomarkers, the ceramide ratios improve the risk stratification in patients with coronary heart disease [8] and are good predictors of CV events, CV mortality, and overall mortality, being thus implemented in clinics [8,12]. Besides ceramides, additional lipid families (i.e., diacylglycerols, sphingomyelins, lysophosphatidylcholines) play an important role as signaling molecules, although their specific role as predictive biomarkers of CVD has been less explored in clinical studies compared to that of ceramides.

Moderate wine consumption is known to be associated with beneficial health effects, especially in CVD but also in neurodegenerative diseases [13]. Wine contains ethanol (whose abuse is undoubtedly harmful) that in low amounts exerts protective effects in terms of total mortality [14]. In addition to alcohol, wine also contains phenolic compounds, which are known to exert beneficial health effects. One of these phenolic compounds is tyrosol (TYR), an antioxidant that is present in relevant amounts in wine (several mg/L range) [15]. Following absorption, the TYR in wine is converted into hydroxytyrosol (HT, a potent cardioprotective antioxidant) via different isoforms from cytochrome P450 (CYP) [16]. In a previous report from a randomized clinical trial in individuals at CV risk, we found that the supplementation of wine with TYR resulted in the endogenous bioactivation of TYR into HT and in cardioprotective effects (i.e., improved endothelial function and increased HDL cholesterol) [16].

In this report, we evaluate the effects of the administration of white wine (WW) and TYR on circulating levels of ceramides and their related ratios in humans at high CV risk. Additionally, we explored the effect of WW and TYR administration on additional families of circulating lipids (i.e., monoacylglycerols (MAGs), diacylglycerols (DAGs), lysophosphatidylcholines (LPCs), sphingomyelin (SM), and sphingosine-1-phosphate (S1P)). We hypothesized that the administration of white wine and Tyr would alter the blood lipid profile which, in turn, will provide insight on the cardioprotective effects of this intervention.

## 2. Materials and Methods

### 2.1. Subjects

Participants in the study were aged from 50 to 80 with at least three of the following major cardiovascular risk factors: smoking, hypercholesterolemia (low-density lipoprotein cholesterol (LDL-c) > 130 mg/dL or under lipid lowering medication), overweight/obesity (BMI > 25 kg/m^2^), hypertension (systolic and diastolic blood pressure > 90 and 140 mmHg, respectively, or under hypotensive treatment check), low HDL (<40 mg/dL in men or <50 mg/dL in women), family history of premature coronary heart disease, and/or type 2 diabetes. Additional inclusion/exclusion criteria are outlined elsewhere [17]. Written informed consent to participate was obtained prior to any study-related procedure. Before the beginning of the study, participants underwent a complete medical examination to exclude concomitant medical conditions.

### 2.2. Study Protocol

The design of the clinical study has been previously described [16]. Briefly, the present trial followed a randomized controlled crossover design with three different interventions: control (water ad libitum), white wine (WW), and WW enriched with a capsule of TYR (WW + TYR). Each intervention had a 4-week duration and was preceded by a 3-week washout period. During the study, participants followed a low-phenolic diet and avoided all alcohol consumption other than the WW provided in the study. WW was consumed together with a meal. In the WW intervention, female participants consumed one glass of WW equivalent to 135 mL of WW, 13.5 g of alcohol, 1.4 mg of TYR, and 0.2 mg of HT. Male participants consumed two glasses of WW distributed between two different meals equivalent to 270 mL of WW, 27 g of alcohol, 2.8 mg of TYR, and 0.4 mg of HT. WW + TYR intervention consisted of the equivalent amount of WW complemented with a capsule of 25 mg TYR. Thus, females consumed 25 mg of TYR and male participants consumed 50 mg of TYR.

The study was conducted in accordance with the Helsinki Declaration and approved by the local Ethical Committee (CEIm-Parc de Salut Mar) and registered in the ClinicalTrials.gov (accessed on 20 October 2021) database (NCT02783989). Informed consent was obtained from all subjects involved in the study.

### 2.3. Cardiovascular Biomarker Assessments

Endothelial function was measured in all participants at the beginning and at the end of each intervention period, monitoring the reactive hyperemia index (RHI) and the augmentation index (AI) measured with EndoPAT 2000 (Itamar Medical Inc., Caesarea, Israel). Biomarkers associated with cardiovascular health such as total cholesterol, HDL-c, antithrombin III (ATIII), and D-dimer (DD) were measured by automated enzymatic methods. LDL cholesterol was calculated by the Friedewald formula whenever triglycerides were inferior to 300 mg/dL. Serum high-sensitivity C reactive protein (hs-CRP), ATIII, and DD were determined by immunoturbidimetry (Horiba, Montpellier, France; Spinreact, Girona, Spain). Homocysteine (Hcy) in plasma was measured by gas chromatography-mass spectrometry (GC-MS) after liquid–liquid extraction. Oxidized LDL (oxLDL), endothelin 1 (ET1), and plasminogen activator inhibitor-1 (PAI-1) were measured in plasma by ELISA (Mercodia, Uppsala, Sweden; Invitrogen, CA, USA; and Affymetrix, CA, USA, respectively).

### 2.4. Plasma Sample for Analysis

Blood samples were obtained at the beginning and at the end of each intervention period at fasting conditions. Venous blood samples were collected in tubes containing EDTA and centrifuged at 1700× *g* for 15 min at 4 °C to obtain plasma. All samples were stored at −80 °C until analysis.

### 2.5. Lipidomic Profile Analysis

Circulating levels of ceramides (Cer) and additional lipids (MAGs, DAGs, HexCers, LPCs, SM, and S1P) in human plasma were determined by liquid chromatography coupled to a tandem mass spectrometry (LC–MS/MS) system, as previously described [18] with slight modifications. Briefly, 10 µL of plasma sample was spiked with 100 µL of methanolic ice-cold internal standard solution containing a mixture of 11 deuterated compounds supplied by Avanti Polar Lipids (see Table A1 for further details). An additional 100 µL of ice-cold methanolic solution was added. After vortexing and centrifugation (5 min, 3500 rpm, 4 °C), the supernatant was transferred to an HPLC vial and 5 µL was injected into the LC-MS/MS system.

The chromatographic separation of the lipid species was performed using an Acquity UPLC instrument (Waters Associates, Milford, MA, USA) operated using the MassLynx 4.1 software. The LC system was equipped with an Acquity UPLC^®^ (BEH C18, 1.7 µm, 2.1 × 100 mm) column (Waters Associates). The flow rate was 0.3 mL/min and the temperature of the column was set at 55 °C. An isocratic method was used, with a solution of 1 mM of ammonium formate (NH_4_HCOO) and 0.01% HCOOH in methanol as the mobile phase solvent. The total run-time was 5 min. The detection of the ammonium adducts ([M + NH_4_]^+^ in the case of diacylglycerols) and protonated adducts ([M + H]^+^ for the other lipid species) was performed with a triple quadrupole (Xevo TQS-Micro MS, Waters) mass spectrometer equipped with an orthogonal Z-spray electrospray ionization source (ESI) operated in the positive ion mode. The monitoring and quantification of the lipids was performed in the MRM mode using two different 5-min acquisition methods as detailed in Table A1. The concentrations of ceramides were calculated by using external calibration curves with authentic standards. Given that there are no commercially available standards for all the lipid species that were assessed in this study, the results are expressed as a relative ratio and were calculated through dividing the peak area of the analyte by the peak area of the corresponding deuterated internal standard as specified in Table A1.

### 2.6. Statistical Analysis

Statistical analyses were performed with R version 3.0.2 and the R packages used were “multcomp”, “nlme”, “ggplot2”, and “corrplot”. The normality of continuous variables was assessed by normal probability plots. Intra-treatment comparisons were assessed between the baseline and final values of each specific intervention using a Student’s t-test for paired samples. Differences between groups were assessed by an independent t-test. Comparisons among treatments were made first calculating the change produced by the treatment (Δ: final vs. baseline values) and then comparing the Δ using an ANOVA for repeated measures adjusted for by sex, age, BMI, smoking status, and statin treatment. The Pearson correlation coefficient was calculated to evaluate the existence of possible linear associations. Significance was set as *p* < 0.05.

## 3. Results

### 3.1. Baseline Characteristics

A total of 33 volunteers successfully completed the study: 12 women (36.4%) and 21 men (63.6%). The mean age of participants was (SD) 65.3 (6.2) years and the body mass index (BMI) was 32.6 (4.2). A total of 6 volunteers (18.2%) were smokers and 17 volunteers (51.5%) were under statin therapy. Other baseline characteristics are outlined in Boronat et al. 2019 [17].

### 3.2. Baseline Lipidomic Profile

Table A2 outlines the lipidomic profile at the baseline of the interventions. The interaction with potential confounders such as age, sex, smoking, BMI, and statin treatment use was studied at the baseline. Significant statistical differences were found between sexes in MAG 18:2 (mean (SD), 0.22 (0.09) in men compared to 0.14 (0.05) in women), MAG 18:1 (1.27 (0.60) in men compared to 0.81 (0.36) in women), DAG 18:1 18:1 (67.5 (26.8) in men compared to 48.0 (13.3) in women), and Cer C18:0 (0.83 (0.24) in men compared to 1.13 (0.35) in women). Additionally, significant differences were found between non smokers and smokers in the following lipids: DAG 18:0 18:2 (3.09 (1.46) vs. 4.65 (1.84), respectively), LPC 16:0 (1238 (418) vs. 1703 (790), respectively), ceramides Cer C14:0 (0.07 (0.03) vs. 0.10 (0.04), respectively), Cer C16:0 (1.20 (0.35) vs. 1.54 (0.43), respectively), Cer C22:0 (2.21 (0.63) vs. 2.85 (0.67), respectively), and Cer C24:0 (9.41 (2.41) vs. 11.70 (2.24), and S1P (0.38 (0.08) vs. 0.53 (0.15), respectively). Likewise, participants under statin treatment displayed higher levels of DAG 18:0 20:4 compared to untreated participants (1.42 (0.76) vs. 0.84 (0.52)). Baseline Cer levels were also affected by statins treatment, exhibiting lower levels in treated participants compared to untreated participants in the following ceramides: Cer C16:0 (1.09 (0.27) vs. 1.42 (0.40), respectively), Cer C18:0 (0.81 (0.22) vs. 1.06 (0.35), respectively), Cer C22:0 (2.00 (0.56) vs. 2.62 (0.64), respectively), and Cer C24:0 (8.90 (2.22) vs. 10.7 (2.52), respectively). Finally, a moderate negative correlation was found between age and DAG 18:2 18:2 (−0.43, *p* = 0.013). BMI was positively correlated to Cer C18:0 with a coefficient of 0.42 and *p* = 0.017.

### 3.3. Changes Observed in the Lipidomic Profile following the Clinical Trial Interventions

Table 1 describes the plasmatic responses of the lipids and/or ratios analyzed before and after each intervention. Control intervention did not trigger any significant change in lipidic profile. Following WW intervention, there was a significant increase (final vs. baseline) in DAG 16:0 16:0, DAG 18:1 16:0, DAG 18:0 18:2, and DAG 18:0 18:1 which was not observed following WW + TYR intervention. An increase was observed in DAG 18:0 20:4 following WW and WW + TYR. The mentioned changes in DAG were not significant when compared to the changes observed in the other intervention. In the family of ceramides, WW triggered an increase (final vs. baseline) in Cer C18:0, Cer C22:0, and Cer C24:0. WW + TYR intervention increased the two latter ones. Accordingly, reductions were also observed in ceramides ratios: Cer C16:0/Cer C24:0, Cer C18:0/Cer C24:0, and Cer C24:1/Cer C24:0 following WW + TYR (final vs. baseline). These reductions were significant when compared to changes observed in the control intervention (Figure 1). The WW intervention only produced a decrease of the Cer ratio 16:0/24:0 (final vs. baseline). Table A3 outlines the Δ of all lipid species following each one of the interventions.

### 3.4. Correlation between Lipidomic Profile and Cardiovasuclar Biomarkers at Baseline

Figure 2 shows the correlation matrix of the lipidomic profile at the baseline of the study and baseline concentrations of the measured cardiovascular biomarkers. In general, the analyzed lipids exhibited a high correlation between lipids from the same family, but moderate correlations were also observed between lipids from different families. Augmentation index (AI) was positively related to DAG 16:1 16:1 and S1P and negatively correlated to DAG 18:2 18:2. Total cholesterol, LDL-c, and LDLox concentrations were positively correlated to LPC and ceramides. On the contrary, HDL-c was only correlated to MAG and DAG families. The concentrations of the pro-thrombotic marker DD was positively correlated to DAG 18:0 18:1, ceramide ratio Cer C18:0/Cer C24:0 and SM (d18:1/18:0), and negatively correlated to DAG 18:0 20:4. In line, ATIII was negatively correlated to ceramide C18:0 and ratio Cer C18:0/Cer C24:0. Hcy was correlated to Cer C24:1 and its ratio Cer C24:1/Cer C24:0. Finally, CRP was correlated to DAG 18:2 18:2, Cer C18:0, and its ratio Cer C18:0/Cer C24:0. Interestingly, DAG 18:0 18:1 showed a correlation profile opposed to the one of DAG 18:0 20:4. Globally, the results suggest that DAG 18:2 18:2 was associated with an improvement in endothelial function and arterial stiffness (decrease in AI). On the contrary, DAG 16:1 16:1 and S1P were correlated with endothelial dysfunction. On the other side, there was a general correlation to ceramides, MAG, and DAG with a dysregulation of cholesterol biomarkers (increase in total cholesterol, LDL-c, and LDL ox and decrease in HDL-c). A pro-thrombotic and pro-inflammatory profile was correlated to high levels of DAG 18:0 18:0, Cer C18:0, and Cer C24:1. On the contrary, DAG 18:2 18:2 and DAG 18:0 20:4 were correlated to a regulation of biomarkers associated with an anti-thrombotic and anti-inflammatory effect.

### 3.5. Correlation between Changes Observed in the Lipidomic Profile with Changes in Cardiovascular Biomarkers

Figure 3 outlines the correlation matrix between changes observed in the lipidomic profile and changes in CV biomarkers following WW + TYR intervention. Increases in MAG and DAG were negatively related to total HDL-c and LDL-c and ATIII. On the contrary, ceramides were positively related to total HDL-c and LDL-c and negatively associated with ATIII. Increases in Hcy were negatively correlated to LPC and ceramides. Finally, changes in the pro-inflammatory molecule CRP were moderately correlated to changes in DAG 16:0 16:0, DAG 18:2 18:2, DAG 18:0 18:2, Cer C16:0, and the ratio Cer C16:0/Cer C24:0.

## 4. Discussion

The present study reports (i) the plasma lipidomic alterations that take place after the administration of WW and WW + TYR in individuals at high CVD risk and (ii) the correlation between the alteration of such plasma lipid species and known CVD risk biomarkers. In a previous study [16], we showed that TYR and its partial biotransformation to HT promoted CV benefits in humans in the context of patterns compatible with dietary wine intake with meals. In this study, we evaluated the effect of dietary doses of WW (a matrix poor in phenolic compounds; WW intervention) vs. the same matrix combined with the dietary antioxidant TYR (WW + TYR intervention) on plasma lipid species with a special focus on changes in circulating ceramides, considered a novel CV biomarker [19].

When we analyzed the plasma levels of a panel of 25 lipid species and ratios, we could observe slight baseline differences depending on sex (as observed in MAG 18:1, MAG 18:2, DAG 18:1 18:1, and Cer C18:0), smoking status (as observed in DAG 18:0 18:2, LPC 16:0, several ceramides, and sphingosine-1-phosphate), statin treatment (which resulted in lower levels of four ceramides), and BMI. Regarding the latter, a strong positive association was found between BMI and Cer C18:0. This observation agrees with the literature, as the concentrations of this ceramide have been reported to be increased in type 2 diabetic subjects and have a positive correlation with BMI [20].

Our results confirm the modulation of the lipidomic profile by dietary interventions. WW intervention increased plasma levels of three ceramides and four DAGs. This effect could be attributed mostly to the alcoholic fraction of WW since its phenolic composition is minor. Parallelly, the supplementation of WW with TYR resulted in a moderate increase of Cer C22:0 and Cer C24:0, and a decrease in the ceramide ratios: Cer C16:0/Cer C24:0, Cer C18:0/Cer C24:0, and Cer C24:1/Cer C24:0. In the case of the DAGs, the alterations following WW + TYR intervention did not reach significance (except for DAG 18:0 20:4), suggesting that the addition of TYR was able to normalize, in part, the WW-induced alterations in plasma DAGs. As expected, the control intervention did not alter the plasma concentrations of the lipid species.

The reduction of ceramides ratios described in the present study is a major finding and of relevance since these ratios are predictors for CV death in individuals with coronary artery disease and are associated with the risk of major CV events in the general population. Indeed, recent studies have used circulating ceramide concentrations to calculate a risk predictor score (the Coronary Event Risk Test (CERT)) and it has been found that it is a powerful predictor of CV mortality in CVD patients [12,21]. Initially described by Laaksonen et al., the ceramide risk score uses these three ratios and the concentrations of Cer C16:0, Cer C18:0, and Cer C24:0 [10]. Currently, the ceramide risk score is used by Mayo Clinic to calculate the risk of myocardial infarction, acute coronary syndromes, and mortality within 1 to 5 years [20]. Our results indicate that the reductions in ceramides ratios are only observable following the supplementation of WW with TYR. Therefore, the biological activity could be attributed to TYR and its following bioactivation into HT. To the best of our knowledge, our study represents one of the few works that shows an in vivo effect of dietary phenolic compounds into ceramides ratios, enlarging the body of evidence of the cardioprotective effects of TYR- and HT-rich foods (i.e. extra virgin olive oil and wine).

Given that in a previous study we had evaluated the cardioprotective effects of these interventions, we decided to explore the existing correlation between the plasma lipidomic profile and CVD risk biomarkers in individuals at high CV risk. As abovementioned, WW + TYR intervention was associated with an improvement in several CV risk factors. Our exploratory analysis allowed us to identify associations between ceramides circulating levels and the mentioned CV improvements. An interesting finding of this study was the strong positive correlation between all basal circulating levels of ceramides and total cholesterol, LDL-c and oxLDL, whereas HDL-c correlated only with Cer 18:0. Additionally, Cer ratio Cer C18:0/Cer C24:0 was negatively correlated with ATIII, a pleiotropic molecule with anti-coagulation and anti-inflammatory effects. Finally, another interesting correlation was observed between the pro-inflammatory biomarker Hcy and ceramides ratios. Overall, our results suggest a potential link between ceramides alterations, the modulation of important parameters key to the pathogenesis of CVD and the biological activities of TYR and HT.

Additionally, the correlation study between the lipidomic profile at baseline and CV biomarkers revealed that DAG 18:2 18:2 was associated with an improved endothelial function and arterial stiffness markers. DAG 18:2 18:2 is a diacylglycerol containing two molecules of linoleic acid esterified with a molecule of glycerol. Linoleic acid is inversely associated with coronary heart disease risk in a dose-dependent manner, according to prospective observational studies [22]. Thus, our finding agrees with current recommendations to replace saturated fat with polyunsaturated fat for prevention of CVD [22]. Conversely, DAG 18:0 20:4, which showed a significant increase following WW + TYR, was correlated to AI at baseline. On the contrary, basal DAG 16:1 16:1 was associated with endothelial dysfunction in our study population. DAG 16:1 16:1 is a diacylglycerol containing two molecules of palmitoleic acid esterified with a molecule of glycerol. Palmitoleic acid is a known marker of de novo lipogenesis, and it correlates to endothelial dysfunction since it is usually released to restore a compromised vascular functionality by improving the bioavailability of endothelial nitric oxide synthase (eNOS) [23]. In the context of DAG, the importance of studying specific DAG moieties has recently been highlighted, due to the divergent or even opposite effects depending on the specific DAG reported [24]. This fact is observable in our baseline correlation study in which opposite correlation patterns are observed between DAG 18:0 18:1 and DAG 18:0 20:4, emphasizing the importance of studying individual DAGs.

White wine was used as a matrix to administer TYR, not only because it is a common constituent of wine, but also because its bioavailability is dependent on the hydroalcoholic matrix provided by wine [25]. If TYR is not administered in an appropriate matrix, its absorption decreases dramatically. Therefore, this study is not specifically designed to evaluate the effects of ethanol on CV risk factors, but to evaluate the contribution of a given phenolic compound TYR (and its endogenous biotransformation in HT), administered in doses superior to those expected in a glass of wine (the dose is equivalent to the content of one liter of wine rich in TYR). For the previously mentioned reasons, an intervention with only TYR (without an appropriate matrix) was not envisaged in the experimental design of this study.

This study has some limitations. Firstly, the number of volunteers recruited is small (especially regarding women), which was impacted by restrictive inclusion criteria, although the crossover design of the study partially counterbalanced this limitation in the population size. However, since this is the first study that evaluates the efficacy of TYR in vivo in modulating cardiovascular parameters, the number of volunteers was deemed sufficient to give an initial answer to our hypotheses. Secondly, a longer observation time would have allowed for the evaluation of the effects of the treatment in the long term, but it would have also affected the volunteers’ compliance and further reduced the number of participants. On the other hand, a limited observation time allowed us to control for every possible lifestyle factor that could have affected the outcome of the intervention. Thirdly, the correlations found between endothelial function parameters and the levels of circulating lipids provide descriptive information but do not establish causality, evidencing the need for additional studies focused on the role of lipids on the development of CVD.

## 5. Conclusions

In this study, we show for the first time that TYR, administered with WW, is effective in modifying the lipidomic profile. The most relevant change is a reduction in the ceramides ratios following WW + TYR intervention. Additionally, the previously described positive effects of WW + TYR on endothelial and CV biomarkers were associated with changes in lipidomic profile. Given the complexity of the data obtained in this research, further molecular investigations will be needed to better understand if the lipid alterations described here are part of the mechanisms involved in the CV positive effects associated with TYR and its in vivo derivative HT. Despite its limitations, this study confirms the relevance of ceramides and other lipids as biomarkers for CVD risk, highlighting different correlations between the concentration of the lipids analyzed, endothelial health, and relevant CV biomarkers.

## Figures and Tables

**Figure 1 antioxidants-10-01679-f001:**
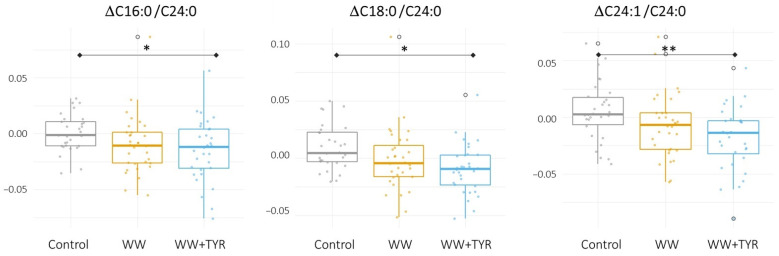
Changes in Δceramides ratio (final vs. baseline) observed following each intervention of the clinical study. Changes in Δceramides ratio (final vs. baseline) observed following each intervention. Statistical differences found between Control and WW + TYR interventions in the ANOVA repeated measures adjusted by sex, age, BMI, smoking status, and statin treatment. * *p* value < 0.05; ** *p* value < 0.001.

**Figure 2 antioxidants-10-01679-f002:**
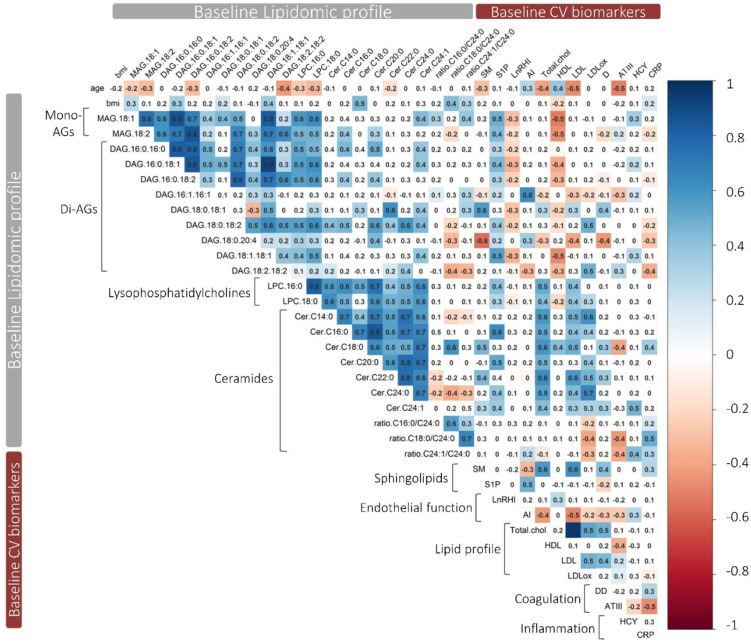
Correlation matrix of the lipidomic profile with CV biomarkers at the baseline of the study. Pearson’s correlation matrix of the lipidomics with CV biomarkers (*n* = 33) at baseline. Blank squares represent correlations with a *p*-value > 0.05. Positive correlations are shown in blue and negative correlations are shown in red. Abbreviations: AGs, acylglycerols; CV, cardiovascular; SM, sphingomyelin; S1P, sphingosine 1 phosphate; LnRHI, logarithm of reactive hyperemia index; AI, augmentation index; Total chol, total cholesterol; DD, D-dimer; HCY, homocysteine; CRP, C-reactive protein.

**Figure 3 antioxidants-10-01679-f003:**
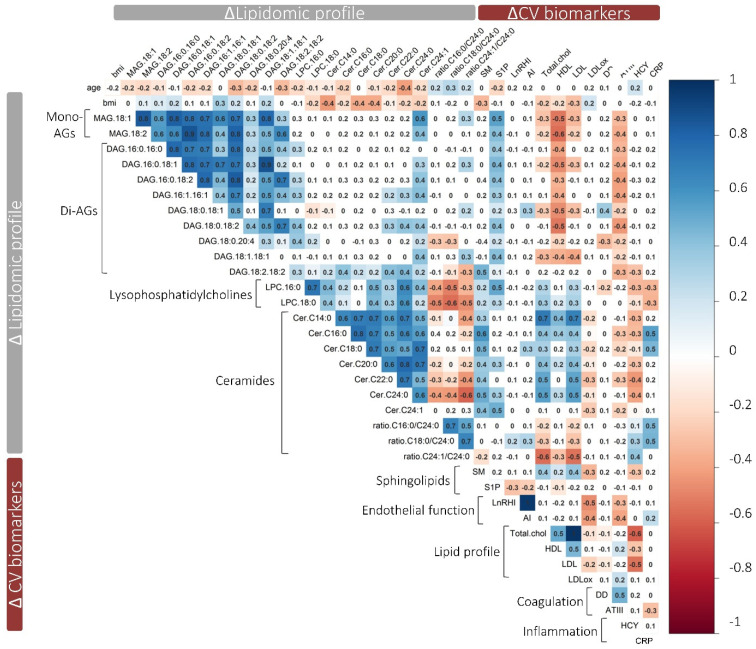
Correlation matrix of the changes observed in the lipidomic profile after WW + TYR intervention with the changes observed in CV biomarkers. Pearson’s correlation matrix of the Δ of the lipidomic profile (final vs. baseline) with the Δ observed in CV biomarkers (*n* = 33) at the end of WW + TYR intervention. Blank squares represent correlations with a *p*-value > 0.05. Positive correlations are shown in blue and negative correlations are shown in red. Abbreviations: AGs, acylglycerols, CV, cardiovascular, SM, sphingomyelin; S1P, sphingosine 1 phosphate. LnRHI, logarithm of reactive hyperemia index; AI, augmentation index; Total chol, total cholesterol; DD, D-dimer; HCY, homocysteine; CRP, C-reactive protein.

**Table 1 antioxidants-10-01679-t001:** Effects in the lipidomic profile of the 4-week control, WW, and WW + TYR intervention.

Lipid	Control		WW		WW + TYR	
	Baseline	Final	Baseline	Final	Baseline	Final
MAG 18:1	0.96 (0.49)	0.96 (0.55)	1.03 (0.52)	1.09 (0.72)	1.07 (0.68)	0.89 (0.35)
MAG 18:2	0.18 (0.08)	0.20 (0.18)	0.18 (0.08)	0.19 (0.12)	0.19 (0.10)	0.16 (0.07)
DAG 16:1 16:1	0.52 (0.21)	0.63 (0.34)	0.55 (0.28)	0.66 (0.38)	0.53 (0.28)	0.61 (0.28)
DAG 16:0 16:0	5.07 (2.23)	5.73 (4.88)	4.92 (2.30)	6.17 (3.75) *	5.11 (2.68)	5.53 (2.70)
DAG 16:0 18:2	8.01 (3.36)	8.82 (5.60)	8.29 (3.62)	9.46 (5.63)	8.95 (4.97)	8.39 (3.54)
DAG 16:0 18:1	27.91 (11.44)	31.65 (18.72)	28.28 (11.84)	32.88 (17.41) *	29.94 (15.12)	29.68 (10.45)
DAG 18:2 18:2	1.09 (0.63)	1.11 (0.74)	1.09 (0.60)	1.11 (0.75)	1.03 (0.51)	0.99 (0.57)
DAG 18:0 18:2	2.85 (1.26)	3.51 (2.87)	2.89 (1.11)	3.55 (1.94) *	2.95 (1.42)	3.06 (1.20)
DAG 18:1 18:1	59.46 (23.62)	66.22 (30.81)	59.85 (22.70)	64.58 (27.87)	64.67 (30.19)	59.19 (18.55)
DAG 18:0 18:1	3.70 (1.72)	4.62 (3.81)	3.66 (1.53)	4.51 (2.52) *	3.62 (1.89)	4.05 (1.50)
DAG 18:0 20:4	1.14 (0.58)	1.38 (0.89)	1.10 (0.74)	1.48 (1.05) *	1.24 (0.66)	1.43 (0.66) *
LPC 16:0	1302.00 (478.38)	1375.30 (470.45)	1337.74 (454.98)	1486.43 (667.47)	1385.23 (519.51)	1435.84 (417.02)
LPC 18:0	578.59 (289.26)	600.51 (282.72)	665.33 (261.74)	650.86 (324.17)	647.24 (275.40)	653.17 (274.35)
Cer C14:0	0.08 (0.03)	0.09 (0.05)	0.09 (0.04)	0.09 (0.04)	0.08 (0.04)	0.09 (0.04)
Cer C16:0	1.24 (0.29)	1.28 (0.37)	1.25 (0.33)	1.30 (0.35)	1.28 (0.38)	1.28 (0.30)
Cer C18:0	0.94 (0.36)	1.00 (0.37)	0.94 (0.34)	1.04 (0.35) *	0.98 (0.35)	0.97 (0.27)
Cer C20:0	3.91 (1.21)	4.26 (1.42)	3.83 (1.20)	4.46 (1.63)	4.13 (1.41)	4.31 (1.15)
Cer C22:0	2.57 (0.80)	2.72 (0.76)	2.55 (0.77)	2.89 (0.88) *	2.53 (0.74)	2.79 (0.69) *
Cer C24:0	9.75 (2.35)	10.11 (2.40)	10.03 (2.69)	11.27 (2.90) *	10.03 (2.73)	11.18 (2.36) *
Cer C24:1	15.45 (4.20)	16.10 (4.27)	14.92 (4.44)	16.16 (4.72)	16.04 (5.06)	15.79 (3.73)
Ratio C16:0/C24:0	0.13 (0.02)	0.13 (0.03)	0.13 (0.03)	0.12 (0.03) *	0.13 (0.04)	0.12 (0.03) *
Ratio C18:0/C24:0	0.10 (0.03)	0.10 (0.03)	0.10 (0.03)	0.10 (0.03)	0.10 (0.03)	0.09 (0.03) *
Ratio C24:1/C24:0	1.60 (0.33)	1.62 (0.36)	1.53 (0.40)	1.46 (0.32)	1.62 (0.32)	1.43 (0.28) **
SM (d18:1/18:0)	12.38 (3.88)	11.94 (3.50)	13.27 (4.42)	12.41 (3.46)	12.48 (3.15)	11.72 (2.45)
S1P	0.38 (0.08)	0.36 (0.08)	0.37 (0.09)	0.38 (0.11)	0.36 (0.12)	0.37 (0.10)

The results are expressed as relative ratios and were calculated through dividing the peak area of the analyte by the peak area of the corresponding deuterated internal standard as specified in Table A1. Results are shown as mean (SD); * *p* value < 0.05 compared to baseline of the intervention; ** *p* value < 0.001 compared to baseline of the intervention. Abbreviations: Cer, ceramide; DAG, diacylglycerols; MAG, monoacylglycerols; SM, sphingomyelin; S1P, sphingosine-1-phosphate.

## Data Availability

Additional data can be found in Boronat, A.; Mateus J.; Soldevila-Domenech, N.; Guerra, M.; Rodríguez-Morató, J.; Varon, C.; Muñoz, D.; Barbosa, F.; Morales, J.C.; Gaedigk, A.; Langohr, K.; Covas, M.I.; Pérez-Mañá, C.; Fitó, M.; Tyndale, R.F.; de la Torre, R. Data on the endogenous conversion of tyrosol into hydroxytyrosol in humans. Data brief. 2019 Nov 12;27:104787, doi:10.1016/j.dib.2019.104787. PMID: 31788516; PMCID: PMC6880089.

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
