# Peer review of "Effects of Wine and Tyrosol on the Lipid Metabolic Profile of Subjects at Risk of Cardiovascular Disease: Potential Cardioprotective Role of Ceramides"

_antioxidants, 2021, doi:10.3390/antiox10111679_

Round 1

Reviewer 1 Report

Comments to the paper entitle « Effects of wine and Tyrosol on the lipid metabolic profile of subjects at risk of cardiovascular disease: potential cardioprotective role of ceramides”

In this manuscript the authors have evaluated the effects of white wine and tyrosol on circulating levels of ceramides in a human population study of high CV risk. The same evaluation was also extended to other lipid species such as TAGs, MAGs, DAGs, SM and some phospholipids by using a targeted lipidomics approach. The aim of this work is oh high interest but some critical points must be addressed before considering it for a publication on Antioxidants.

Major comments:

  • A major concern of this manuscript is the experimental design. As described in the present manuscript this study consist of three different interventions: Control, white wine, and white wine + tyrosol with the main objective of evaluating the effect of white wine and tyrosol on circulating levels of ceramides. The benefits as a cardiprotective of tyrosol and moderate wine consumption were described by the authors in a previous publication (Boronat, A., et al., Cardiovascular benefits of tyrosol and its endogenous conversion into hydroxytyrosol in humans. A randomized, 492 controlled trial. Free Radic Biol Med, 2019. 143: p. 471-481). However, it is not clear if the positive effect observed is only associated to tyrosol or to the combination of white wine and tyrosol. An additional intervention where the same participants are supplemented with Tyrosol only would help to understand if tyrosol alone is also cardioprotective and as a consequence the ceramides ratios used as a new markers of cardiovascular risk effects decrease as well.

  • Some confounding factors like sex, BMI, age and smoking status has been considered for statistical analysis. But other confounding factors such as medication should be consider. Are the participants under statin treatment?

  • Regarding the analytical methodology, the authors has applied a targeted lipidomics analysis. There is a very good justification about targeting ceramides since they have been recently used as a new biomarkers of CVD. In fact Mayo clinic has already established a Risk score system based on the ratios of several ceramides measured in plasma. https://www.mayocliniclabs.com/test-catalog/Clinical+and+Interpretive/606777. I do suggest to add a discussion about the ratios observed in comparison with the score calculation used at Mayo clinic.

  • However, there is any justification about the other lipid species measured together with Ceramides. The chromatographic separation used in this work is not the most suitable for the analysis of other lipid species. So, I do suggest to remove all the results and discussion about other lipid species and focus the publication only on ceramides analysis. The limited number of lipid species measured is not enough to have a comprehensive lipidomic profile. Moreover, the data is not quantitative and in clinical studies this is very critical for cross comparison between different clinical cohorts.

Author Response

Reviewer 1

In this manuscript the authors have evaluated the effects of white wine and tyrosol on circulating levels of ceramides in a human population study of high CV risk. The same evaluation was also extended to other lipid species such as TAGs, MAGs, DAGs, SM and some phospholipids by using a targeted lipidomics approach. The aim of this work is oh high interest but some critical points must be addressed before considering it for a publication on Antioxidants.

 Major comments:

  • A major concern of this manuscript is the experimental design. As described in the present manuscript this study consist of three different interventions: Control, white wine, and white wine + tyrosol with the main objective of evaluating the effect of white wine and tyrosol on circulating levels of ceramides. The benefits as a cardiprotective of tyrosol and moderate wine consumption were described by the authors in a previous publication (Boronat, A., et al., Cardiovascular benefits of tyrosol and its endogenous conversion into hydroxytyrosol in humans. A randomized, 492 controlled trial. Free Radic Biol Med, 2019. 143: p. 471-481). However, it is not clear if the positive effect observed is only associated to tyrosol or to the combination of white wine and tyrosol. An additional intervention where the same participants are supplemented with Tyrosol only would help to understand if tyrosol alone is also cardioprotective and as a consequence the ceramides ratios used as a new markers of cardiovascular risk effects decrease as well.

We thank the reviewer for this comment. As the reviewer points out, the experimental design of our study consisted of 3 different interventions: (1) control (water ad libitum), (2) white wine, and (3) white wine enriched with a capsule of tyrosol. Previous studies from us [1,2] indicate that the bioavailability of tyrosol is extremely dependent on the matrix. A hydroalcoholic matrix (like the one present in wine) is a key factor that enables the absorption of tyrosol, whereas the administration of tyrosol in the form of a capsule and without an appropriate matrix, would have resulted in an extremely low absorption. Thus, we chose a white wine matrix since it is very poor in phenolic compounds (intervention 2) and then we had a different intervention with the same matrix but supplemented with tyrosol (intervention 3). By comparing intervention number 2 and intervention number 3, we can evaluate which are the effects of tyrosol, when this compound is administered in a white wine matrix. To clarify the rationale behind our experimental design, we have modified the following paragraph of the discussion as follows:

“White wine was used as a matrix to administer TYR, not only because is a common constituent of wine, but also because its bioavailability is dependent on the hydroalcoholic matrix provided by wine [33]. If TYR is not administered in an appropriate matrix, its absorption decreases dramatically. Therefore, this study is not specifically designed to evaluate the effects of ethanol on CV risk factors, but to evaluate the contribution of a given phenolic compound TYR (and its endogenous biotransformation in HT), administered in doses superior to those expected in a glass of wine (the dose is equivalent to the content of one liter of wine rich in TYR). For the previously mentioned reasons, an intervention with only TYR (without an appropriate matrix) was not envisaged in the experimental design of this study”.

[1] Pérez-Mañá C, Farré M, Rodríguez-Morató J, Papaseit E, Pujadas M, Fitó M, Robledo P, Covas MI, Cheynier V, Meudec E, Escudier JL, de la Torre R. Moderate consumption of wine, through both its phenolic compounds and alcohol content, promotes hydroxytyrosol endogenous generation in humans. A randomized controlled trial. Mol Nutr Food Res. 2015 Jun;59(6):1213-6. doi: 10.1002/mnfr.201400842. Epub 2015 Apr 27. PMID: 25712532.

[2] Rodríguez-Morató J, Boronat A, Kotronoulas A, Pujadas M, Pastor A, Olesti E, Pérez-Mañá C, Khymenets O, Fitó M, Farré M, de la Torre R. Metabolic disposition and biological significance of simple phenols of dietary origin: hydroxytyrosol and tyrosol. Drug Metab Rev. 2016 May;48(2):218-36. doi: 10.1080/03602532.2016.1179754. Epub 2016 May 17. PMID: 27186796.

  • Some confounding factors like sex, BMI, age and smoking status has been considered for statistical analysis. But other confounding factors such as medication should be consider. Are the participants under statin treatment?

We thank the reviewer for this comment, which has offered us the possibility to further investigate the potential confounding effects of statin treatment on the results of this study. As suggested by the reviewer, an in agreement with the literature [3], statin treatment affects the levels of ceramides. At the baseline, the circulating concentrations of 4 ceramides (Cer C16:0, Cer C18:0, Cer C22:0 and Cer C24:0) were lower in patients on statin treatment vs untreated patients. However, the ceramide ratios do not present differences between groups. Given these results, we have now added the statin treatment as a covariable and we have repeated all the statistical analysis. All the corresponding new p values have been added to the new version of the manuscript (see new supplementary table A2). After this modification, none of the major conclusions of the study are altered since all the statistically significant observations are maintained and the parameters which were non-significant before the modification keep being non-significant. Also, the following sentences have been added to the manuscript:

  • Material and methods. Subheading 2.6. Statistical analysis:

“Differences between groups were assessed by an independent t-test. Comparisons among treatments were made first calculating the change produced by the treatment (D: final vs baseline values) and then comparing the D using an ANOVA for repeated measures adjusted for by sex, age, BMI, smoking status and statin treatment.”

  • Suheading 3.1. Baseline characteristics

“Mean age of participants was (SD) of 65.3 (6.2) years and the body mass index (BMI) was 32.6 (4.2). A total of 6 volunteers (18.2%) were smokers and 17 volunteers (51.5%) were under statin therapy.”

  • Subheading 3.2. Baseline lipidomic profile.

“Likewise, participants under statin treatment had higher levels of DAG 18:0 20:4 compared to untreated participants (1.42 (0.76) vs 0.84 (0.52)). Baseline Cer levels were also affected by statins treatment, exhibiting lower levels in treated participants compared to untreated participants in the following Cer: Cer C16:0 (1.09 (0.27) vs 1.42 (0.40), respectively), Cer C18:0 (0.81 (0.22) vs 1.06 (0.35), respectively), Cer C22:0 (2.00 (0.56) vs 2.62 (0.64), respectively) and Cer C24:0 (8.90 (2.22) vs 10.7 (2.52), respectively).”

[3] Han JS, Kim K, Jung Y, Lee JH, Namgung J, Lee HY, Suh J, Hwang GS, Lee SH. Metabolic Alterations Associated with Atorvastatin/Fenofibric Acid Combination in Patients with Atherogenic Dyslipidaemia: A Randomized Trial for Comparison with Escalated-Dose Atorvastatin. Sci Rep. 2018 Oct 2;8(1):14642. doi: 10.1038/s41598-018-33058-x. PMID: 30279504; PMCID: PMC6168550.

  • Regarding the analytical methodology, the authors has applied a targeted lipidomics analysis. There is a very good justification about targeting ceramides since they have been recently used as a new biomarkers of CVD. In fact Mayo clinic has already established a Risk score system based on the ratios of several ceramides measured in plasma. https://www.mayocliniclabs.com/test-catalog/Clinical+and+Interpretive/606777. I do suggest to add a discussion about the ratios observed in comparison with the score calculation used at Mayo clinic.

As mentioned by the reviewer, Laaksonen and colleagues [4] developed a ceramide risk score based on the values of Cer C16:0, Cer C18:0, Cer C24:1, and the corresponding ratios with Cer C24:0. The calculation adds 1 point for every variable that is in the 3rd quartile and 2 points for every variable that is in the 4th quartile. As the specific values for the quartiles are not indicated in that study, we have not been able to replicate that calculation. However, following the reviewer’s advice, we have now added the following paragraph to the discussion about the ratios observed in our study and the score calculation used at Mayo Clinic.

“In the case of ceramides, the WW+TYR intervention resulted in a moderate decrease in the ratios of 3 ceramide ratios (i.e. Cer C16:0/Cer C24:0, Cer C18:0/Cer C24:0 and Cer C24:1/Cer C24:0), when compared to the control intervention. This fact is a major finding of this study and is relevant since these ratios are predictors for CV death in coronary artery disease individuals and are associated with the risk of major CV events in general population. Indeed, recent studies have used circulating ceramide concentrations to calculate a risk predictor score (the Coronary Event Risk Test (CERT)) and it has been found that it is a powerful predictor of CV mortality in CVD patients [12, 20]. Initially described by Laaksonen et al, the ceramide risk score uses these 3 ratios and the concentrations of Cer C16:0, Cer C18:0 and Cer C24:0 [4]. Currently, the ceramide risk score is used by Mayo Clinic to calculate the risk of myocardial infarction, acute coronary syndromes, and mortality within 1 to 5 years [5].”

[4] Laaksonen R, Ekroos K, Sysi-Aho M, Hilvo M, Vihervaara T, Kauhanen D, Suoniemi M, Hurme R, März W, Scharnagl H, Stojakovic T, Vlachopoulou E, Lokki ML, Nieminen MS, Klingenberg R, Matter CM, Hornemann T, Jüni P, Rodondi N, Räber L, Windecker S, Gencer B, Pedersen ER, Tell GS, Nygård O, Mach F, Sinisalo J, Lüscher TF. Plasma ceramides predict cardiovascular death in patients with stable coronary artery disease and acute coronary syndromes beyond LDL-cholesterol. Eur Heart J. 2016 Jul 1;37(25):1967-76. doi: 10.1093/eurheartj/ehw148. Epub 2016 Apr 28. PMID: 27125947; PMCID: PMC4929378.

[5] https://www.mayocliniclabs.com/test-catalog/Clinical+and+Interpretive/606777

  • However, there is any justification about the other lipid species measured together with Ceramides. The chromatographic separation used in this work is not the most suitable for the analysis of other lipid species. So, I do suggest to remove all the results and discussion about other lipid species and focus the publication only on ceramides analysis. The limited number of lipid species measured is not enough to have a comprehensive lipidomic profile. Moreover, the data is not quantitative and in clinical studies this is very critical for cross comparison between different clinical cohorts.

We thank the reviewer for the comment. We acknowledge that the main objective of this work was to evaluate the effect of an intervention consisting of white wine plus tyrosol on the plasma levels of ceramides. However, as an exploratory analysis we also evaluated how other lipids are altered by the treatment and its relationship with well-known cardiovascular biomarkers. To the best of our knowledge, there is limited evidence of the effect of alcohol and dietary polyphenols on individual lipids and the relation of these individual lipids with the measured biomarkers. In our opinion and understanding, the limitations of our study and the exploratory nature of our analysis, our results point out two ideas that go beyond ceramides which are also valuable: (i) alcohol triggers an upregulation of certain lipid species; these upregulation is lessened in the presence of Tyr and HT, and (ii) different correlation patterns between individual lipid species and CV biomarkers are observed, suggesting that the biological activity of lipids should be assessed individually. Additionally, given the comments of reviewer number 2 (most of which were related to the other lipids) we have decided to keep the results of the other lipids but, agreeing with reviewer number 1, we have modified the introduction and the discussion to tune down the results of the other lipids and to emphasize the ceramides results.

Reviewer 2 Report

I don't have major issues to report but I have only some minor issues to be addressed. 

1) Section 2.5 (lipiomic profile analysis): Supporting Information is Table A3 and not A2 (Changes in the lipidomic profile (Δ: final vs baseline) following the 4-week control, WW and WW+TYR interventions)

2) I suggest to review nomenclature of DAG and SM, because it is not easy to interpret in this form. As you are performing a targeted approach, lipid molecules need to be defined in therm of fatty acyl composition.

For example, in the text your refer to SM but then only one species, SM 18:1 was quantified. I'd change SM with the specific SM.Furthermore, m/z 731.5 in positive ion mode is associated to [M+H]+ of SM (d18:0/18:1) and SM (d18:1/18:0). You should specify which one you quantify.

Same issue with DAG. I found quite misleading to read DAG 18:2 . DAG are composed of two fatty acyl chains and both are usually reported in name. A DAG composed of two 18:2 fatty acyl chains is commonly defined as DAG 36:4 or DAG 18:2/18:2 (DAG 18:2 seems like a DAG composed of only one chian, therefore a MAG). For sake of clairty, I suggest to specify the two chains (as done for molecular species composed of two different acyl chains)

line 252: please check sentence "there is a general correlated" doesn't sound well to me. Did you mean " there is a general correlation"?

line 271: add a space between "LDL-c" and "and negatively"

line 299: add space between MAG and 18:2 and DAG and 18:2

line 324: check sentence " this review highlight's".. did you mean " this review highlights"?

line 326: delete ; between study and in

Author Response

I don't have major issues to report but I have only some minor issues to be addressed. 

  • Section 2.5 (lipiomic profile analysis): Supporting Information is Table A3 and not A2 (Changes in the lipidomic profile (Δ: final vs baseline) following the 4-week control, WW and WW+TYR interventions)

We thank the reviewer for the observation. The text has now been corrected accordingly.

  • I suggest to review nomenclature of DAG and SM, because it is not easy to interpret in this form. As you are performing a targeted approach, lipid molecules need to be defined in therm of fatty acyl composition. For example, in the text your refer to SM but then only one species, SM 18:1 was quantified. I'd change SM with the specific SM. Furthermore, m/z 731.5 in positive ion mode is associated to [M+H]+ of SM (d18:0/18:1) and SM (d18:1/18:0). You should specify which one you quantify.

We agree with the reviewer. In order to avoid possible ambiguities in the nomenclature of SM we have now replaced the previous nomenclature of “SM18:1” by “SM (d18:1/18:0)” since it’s the nomenclature recommended by the reviewer and suggested by LipidMaps initiative. The manuscript text, tables and figures have been modified accordingly.

  • Same issue with DAG. I found quite misleading to read DAG 18:2 . DAG are composed of two fatty acyl chains and both are usually reported in name. A DAG composed of two 18:2 fatty acyl chains is commonly defined as DAG 36:4 or DAG 18:2/18:2 (DAG 18:2 seems like a DAG composed of only one chian, therefore a MAG). For sake of clairty, I suggest to specify the two chains (as done for molecular species composed of two different acyl chains)

 In order to clarify this point, we have now specified the 2 fatty acyls chains of every DAG as follows: DAG 16:0 16:0, DAG 16:1 16:1, DAG 18:0 18:0, DAG 18:1 18:1 and DAG 18:2 18:2. The corresponding text, tables and figures have been modified accordingly.

line 252: please check sentence "there is a general correlated" doesn't sound well to me. Did you mean " there is a general correlation"?

The reviewer is right. The manuscript has been modified accordingly.

line 271: add a space between "LDL-c" and "and negatively"

A space has been added.

line 299: add space between MAG and 18:2 and DAG and 18:2

A space has been added.

line 324: check sentence " this review highlight's".. did you mean " this review highlights"?

“Highlight’s” has now been replace by “highlights”.

line 326: delete ; between study and in

Deleted. We appreciate all comments of the reviewers that have enabled us to improve the quality of the manuscript. 

Round 2

Reviewer 1 Report

The authors have replied and modified the manuscript according to the reviewer's suggestions/comments improving the quality of the manuscript.